# Local $K$-Similarity Constraint for Federated Learning with Label Noise

## Abstract

Federated learning on clients with noisy labels is a challenging problem, as such clients can infiltrate the global model, impacting the overall generalizability of the system. Existing methods proposed to handle noisy clients assume that a sufficient number of clients with clean labels are available, which can be leveraged to learn a robust global model while dampening the impact of noisy clients. This assumption fails when a high number of heterogeneous clients contain noisy labels, making the existing approaches ineffective. In such scenarios, it is important to locally regularize the clients before communication with the global model, to ensure the global model isn't corrupted by noisy clients. While pre-trained self-supervised models can be effective for local regularization, existing centralized approaches relying on pretrained initialization are impractical in a federated setting due to the potentially large size of these models, which increases communication costs. In that line, we propose a regularization objective for client models that decouples the pre-trained and classification models by enforcing similarity between close data points within the client. We leverage the representation space of a self-supervised pretrained model to evaluate the closeness among examples. This regularization, when applied with the standard objective function for the downstream task in standard noisy federated settings, significantly improves performance, outperforming existing state-of-the-art federated methods in multiple computer vision and medical image classification benchmarks. Unlike other techniques that rely on self-supervised pretrained initialization, our method does not require the pretrained model and classifier backbone to share the same architecture, making it architecture-agnostic.

## 1 Introduction

Federated learning is a decentralized, privacy-preserving training approach that learns from distributed data across clients to build a global model. Prior works on federated learning have mostly focused on tackling the issues of data heterogeneity and convergence guarantees (Li et al., 2020b; Karimireddy et al., 2020; Li et al., 2021), and assume that client data contains clean labels. However in real-world settings, for instance, in the medical scenario, variability in the expertise of clinicians and the use of automated labeling algorithms to reduce annotation costs can result in frequent label noise (Shi et al., 2024; Khanal et al., 2023). The presence of label noise in client data introduces a unique challenge within the federated learning framework.

Several works have been proposed to mitigate the impact of noisy labels and improve the model's robustness in centralized training setup (Han et al., 2018; Li et al., 2020a; Arazo et al., 2019). However, such Noisy Label Learning (NLL) methods cannot be directly applied to federated scenarios, as these methods often require large amounts of balanced data to train–while clients in federated systems generally have limited and imbalanced data–and are difficult to adapt in a privacy-preserving manner (Xu et al., 2022; Wu et al., 2023). Additionally, the challenge of dealing with label noise in the federated setting is aggravated by the heterogeneity among clients, as each client can have distinct data distribution and inconsistent noise patterns.

Existing Federated Noisy Label Learning (FNLL) methods rely on the presence of a substantial number of clean clients, and use the global model at the server to either regularize the local training or denoise the local client datasets. For instance, FedNoro (Wu et al., 2023) uses KL divergence between the logits generated

by the global and local model to regularize local client training, and uses distance-aware aggregation of local models to decrease the impact of label noise on the global model. Similarly, FedCorr (Xu et al., 2022) first identifies noisy clients and uses the global model trained on clean clients to denoise the noisy samples. These methods rely on the assumption that the global model trained in such a manner is more robust to the local noise. However, when the noise level in the federated system is increased, there will likely not be enough noise-free or low-noise clients to properly warm up the global model, reducing the effectiveness of such approaches. This underscores the necessity of a robust local regularization approach to combat the impact of noisy clients and enhance the resilience of the global model.

Self-supervised models are trained without class labels and can offer a robust starting point for training on limited data (Chen et al., 2020; Caron et al., 2021). Moreover, representations learned in this way are resilient to label noise, enabling robust learning even with noisy annotations (Khanal et al., 2023; Tsouvalas et al., 2024). A straightforward approach to leveraging pre-trained self-supervised learning (SSL) models for robust local regularization is to initialize client encoders with pretrained checkpoints. However, this requires that the client model match the available SSL encoder architecture. Since foundation SSL models are typically very large, this can significantly increase communication costs during federated training.

To address this, we propose a novel decoupled approach for robust local training in federated learning by only relying on the representations of client data, obtained from SSL pretrained model. A key question here is – *How do we most effectively utilize representations obtained from SSL pretrained model for federated learning with label noise?* Towards this, we present a K-Nearest Neighbour contrastive objective, which constrains the representation space of the client model to exhibit a similar local neighborhood as that of the SSL pretrained model. Our approach is motivated by the empirical observation that datasets innately form subclass clusters in representations, obtained from the SSL pretrained model (Figure 2a). Unlike FedLN-AKD, a recent work trained with the objective of directly mimicking the representation of SSL pretrained model (Tsouvalas et al., 2024), our approach relies on SSL representation to only locate immediate neighboring samples, making it generalizable to even out-of-domain scenarios where the pretraining data and client's data differ vastly, as in the scenario of medical datasets. Similarly, in contrast to simply initializing the model from SSL-pretrained weight, our approach is architecture-agnostic, allowing flexible selection of SSL and client models and control over communication costs. Furthermore, as the proposed regularization mechanism is applied only during local training and does not rely on global models, the method is effective even in cases of high but realistic noise levels. We note that while the quality of SSL representations does influence performance, our contribution lies not in selecting the best SSL model, but in designing a more robust framework for utilizing these representations under label noise. Overall, our main contributions can be summarized as follows:

- We propose a novel regularization approach for robust training mechanisms in each client, that preserves locality between the representation space of the pre-trained self-supervised model and classifier.

- We conduct extensive experiments on various noise levels, noise patterns (random label flipping and real-life noise) and multiple datasets–including multiple computer vision and medical image classification benchmarks–that show the effectiveness of our method across various domains when most of the clients are noisy.

## 2 Related Work

**Federated Learning:** Dealing with local client drift (Li et al., 2020b; 2021; Karimireddy et al., 2020), fairness (Li et al., 2019; Ray Chaudhury et al., 2022), improving convergence (Cho et al., 2020; Tang et al., 2022; Zhang et al., 2023b), heterogeneous model architecture (Lin et al., 2020; Li & Wang, 2019) and domain generalization (Nguyen et al., 2022; Zhang et al., 2023a) have been the core focus of most work in federated learning. However, these works assume that client data is clean, which is an impractical assumption in real-world settings.

**Contrastive Objective in Federated Learning:** Recent works have explored the use of contrastive learning objectives to mitigate different challenges associated with federated learning (FL). MOON (Li et al., 2021) uses contrastive learning to limit the effect of statistical heterogeneity. CreamFL (Yu et al., 2023) addresses

the issue of heterogeneous modality distributions in multimodal FL using a contrastive loss that aligns representations across different modalities. FedPCL (Tan et al., 2022) proposes a lightweight framework that leverages multiple pretrained and fixed encoders. It employs a contrastive objective between client-generated representations and aggregated prototypes to improve generalization. FedRCL (Seo et al., 2024) introduces a relaxed supervised contrastive learning objective to improve the transferability of representations and alleviate inconsistent local updates under client data heterogeneity. FedProc (Mu et al., 2023) also utilizes contrastive learning between client representations and global aggregated prototypes to address data heterogeneity. These methods utilize contrastive learning primarily to align representations between clients and the global server model, demonstrating its effectiveness in tackling various forms of client heterogeneity in federated learning. While FedRGL (Li et al., 2024) incorporates graphical contrastive learning in federated learning with label noise, their work is tailored to graphical data, and does not use self-supervised pretrained models, making it not directly comparable to our work.

**Learning with Noisy Labels:** Several works have been proposed in recent years to tackle the problem of learning with noisy labels (Song et al., 2022). Methods range from the design of noise-robust architectures (Lee et al., 2019; Goldberger & Ben-Reuven, 2022), robust regularization (Zhang et al., 2017; Liu et al., 2020), and loss adjustment (Patrini et al., 2017; Wang et al., 2017), to sample selection strategies (Han et al., 2018; Li et al., 2020a; Tan et al., 2021). While several works have been proposed in the centralized NLL setting, federated learning with noisy clients is still in its early phase.

Recently, some works have been proposed to tackle learning with noisy labels in a federated setup. Yang et al. (Yang et al., 2022) proposed an approach where clients share the class centroids with global model in every round to ensure consistent decision boundaries across clients. However, sharing class centroids can lead to privacy concerns, making their approach practically limiting. FedNed (Lu et al., 2024) and RHFL (Fang & Ye, 2022) proposed distillation-based approaches utilizing a public dataset at the server which may not always be available. FedSLR (Jiang et al., 2022) used the delayed memorization effect and utilized mixup augmentation and self-distillation to prevent the client model from overfitting noisy labels. Recent works such as FedCorr (Xu et al., 2022) and FedNoro (Wu et al., 2023) focused on clean and noisy client separation and handled the training process separately. FedCorr warms up a global model by training only on the detected clean clients, which is then used to denoise the noisy clients before normal FedAvg (McMahan et al., 2017) training. Meanwhile, FedNoro utilizes KL divergence between logits of the global model and detects noisy clients to perform regularization. Our work, however, directly regularizes the client model update without using a global model as a proxy to denoise noisy clients. This enables our method to perform best in conditions where the number of noisy clients is dominant. A recent line of work, utilizing SSL pretrained models, has shown promising results. FedLN-AKD (Tsouvalas et al., 2024) employs a representation head to directly replicate the SSL representations using an L1 loss, which may not be optimal for downstream classification. Our approach only enforces the immediate neighbourhood structure to be preserved in the classifier space, which is empirically shown to be more generalizable and significantly more robust in high-noise settings.

## 3 Methodology

### 3.1 Preliminaries

A general federated learning framework consists of $N$ clients, each with its separate training data, $D_n = \{(x_i^n, y_i^n)_{i=1}^{M_n}\}$ , consisting of tuples of input $x_i^n$ and ground truth label $y_i^n$ . The objective is to train a global model using these $N$ clients' data, aiming for good performance on test data. In federated learning, each client trains the global model on its local dataset and sends the updated model weights back to the server. The server aggregates these local models to form an updated global model. Formally, let, $w_t^n$ denote the $n^{\text{th}}$ local client model and $w_t$ denote the global server model at the end of communication round $t$. In each communication round, we randomly select (without replacement) a set, $S_t$ consisting of $n_t$ clients for training. Each selected client is sent a global model and the client trains the global model on its local dataset

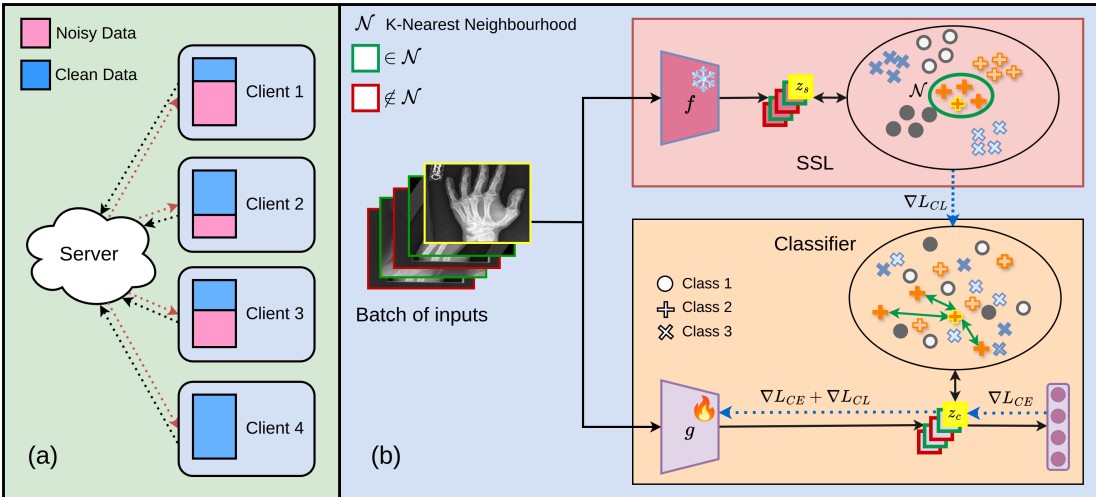

Figure 1: (a) A general federated learning framework with label noise. (b) Illustration of our method: At each client, for a given input image (yellow), we find the $K$ nearest neighbours in the batch (green) in the SSL representation space. The neighbours in the SSL space may be far apart in the classifier space due to the effect of label noise. Thus the representations of $K$-nearest neighbours in the neighbourhood $\mathcal{N}$ are constrained to be closer to the input in the classifier representation space using our local $K$-similarity constraint $L_{CL}$.

and sends it back to the server. The server aggregates the received client model as follows:

$$w_t = \sum_{n \in S_t} \frac{|D_n|}{\sum_{i \in S_t} |D_i|} w_t^n \tag{1}$$

However, in the *presence of label noise*, the training process is affected. Label noise occurs when the ground truth labels $y$ will be replaced by noisy label $\hat{y}$, resulting in the dataset $\hat{D}_n = \{(x_i^n, \hat{y}_i^n)_{i=1}^{M_n}\}$. Training with these noisy labels (or noisy clients) can degrade the performance of the global model, making it crucial to address and mitigate the impact of noise in the federated learning process.

### 3.2 Local $K$-similarity constraint

We propose imposing a local $K$-similarity constraint on clients with noisy labels, utilizing a shared self-supervised pretrained model to anchor representations and mitigate the impact of noisy labels, thereby achieving a robust federated learning system. SSL models, trained without labels, can produce robust representations unaffected by noisy labels and improve generalization in downstream applications. Inputs with similar semantics are grouped in representation and form a compact sub-class structure (Xue et al., 2022). Our main idea is to leverage this local structure of SSL representation to *regularize* the client's representation to enhance the robustness against noisy labels.

Our proposed framework (Figure 1 (b)) consists of a frozen SSL pretrained encoder $f$ shared by all clients, along with the client-specific model, comprising a feature extractor $g^n$ and task-specific head $h^n$ (e.g., a linear classifier). During the local training process in a noisy client $C$, a mini-batch $\mathcal{B} = \{(x_{i_j}^n, \hat{y}_{i_j}^n)_{j=1}^m\}$ contains inputs $X = \{x_{i_1}^n, x_{i_2}^n, \dots x_{i_m}^n\}$ and their noisy labels $\hat{Y} = \{\hat{y}_{i_1}^n, \hat{y}_{i_2}^n, \dots \hat{y}_{i_m}^n\}$. The inputs $X$ are fed into both the SSL encoder encoder and feature extractor (i.e., $f$ and $g^n$) to obtain two embedding sets $f(X) := \{z_{i_1}^s, z_{i_2}^s, \dots z_{i_m}^s\}$, where each $z_{i_j}^s := f(x_{i_j}^n)$ denotes the representations produced by SSL pretrained model, and $g_n(X) := \{z_{i_1}^c, z_{i_2}^c, \dots z_{i_m}^c\}$, where each $z_{i_j}^c := g_n(x_{i_j}^n)$ denotes the client feature vector.

Since our objective is to enforce similarity between semantically similar features of input data points for the client's model (classifier) training, we rely on the representation space of the SSL model and use it

for regularizing the representation space of the client's feature extractor. This is achieved as follows: For each $x_{i_j}^n$ in $\{x_{i_j}^n\}_{j=1}^m$, we consider $\mathcal{N} = \mathcal{N}(x_{i_j}^n, K) \subseteq \{i_k\}_{k=1}^m$ the set of indices of the $K$-nearest neighbors of $x_{i_j}^n$ **in the representation space of the SSL model** (where $K \leq m - 1$), and create $K$ positive pairs $\{(z_{i_j}^c, z_{i_k}^c) \mid i_k \in \mathcal{N}\}$ and the remaining $m - K - 1$ negative pairs $\{(z_{i_j}^c, z_{i_k}^c) \mid i_k \in \mathcal{N}, \ i_k \neq i_{i_j}\}$ **in the representation space of the classifier**. Here, the $K$-nearest neighbors $z_{i_k}^c$ are determined by the Euclidean distance between the normalizations of $z_{i_j}^s$ and $z_{i_k}^s$.

We then apply a similarity constraint for these positive and negative pairs of representations in the client's model (classifier) representation space. To handle all the positive and negative pairs together in the batch, we apply an InfoNCE-based contrastive loss that aims to maximize the similarity between the positive representation pairs and minimize the similarity between the negative representation pairs. The loss function for any anchor representation $z_{i_j}^c$ with its $K$ positive pairs is defined as:

$$\mathcal{L}_{CL,i_j(X;f,g_n)} := -\log \frac{\sum_{i_k \in \mathcal{N}} \exp(\langle z_{i_j}^c, z_{i_k}^c \rangle / \mathcal{T})}{\sum_{l=1}^m \exp(\langle z_{i_j}^c, z_{i_l}^c \rangle / \mathcal{T})} \tag{2}$$

where $\mathcal{T}$ denotes a temperature parameter.

This per-sample local $K$-similarity constraint is combined with the standard cross-entropy loss. It should be noted that noisy clients will have data points with both clean and noisy labels, but identifying such data points upfront may not be possible. The cross-entropy loss incurred from both clean and noisy labels will be used to update the model. However, since the loss from noisy labels can distort the learning process, we augment the local $K$-similarity constraint as a regularizer to the cross-entropy loss to balance and mitigate the effect of the noisy cross-entropy loss, ensuring more robust model updates.

The total per-sample objective is thus expressed as a sum of cross-entropy loss and our local $K$-similarity constraint:

$$\mathcal{L}_{i_j}(x_{i_j}^n; X, \hat{Y}, f, g_n, h_n) := \mathcal{L}_{\text{CE}}(\hat{y}_{i_j}^n, h_n(g_n(x_{i_j}^n))) + \lambda \cdot \mathcal{L}_{\text{CL},i_j}(X; f, g_n) \tag{3}$$

where $\lambda$ is a hyperparameter to weigh the contrastive loss $L_{CL,ij}$, and $\mathcal{L}_{CE}(\hat{y}_{i_j}, h_n(g_n(x_{i_j}^n))$ refers to standard cross-entropy loss between the predicted logit, $y_i = h_n(g_n(x_{i_j}^n))$ and label $\hat{y}_{i_j}$, affecting both $g$ and $h$ sections of the client network. Note that we utilize unnormalized representations to obtain the logits $h_n(g_n(x_{i_j}^n))$. We minimize the average loss of all the samples in the minibatch obtaining the final loss as $\mathcal{L} = \frac{1}{m} \sum_{j=1}^m \mathcal{L}_{i_j}(x_{i_j}^n; X, \hat{Y}, f, g_n, h_n)$. . We refer to Algorithm 1 for an overview of the steps.

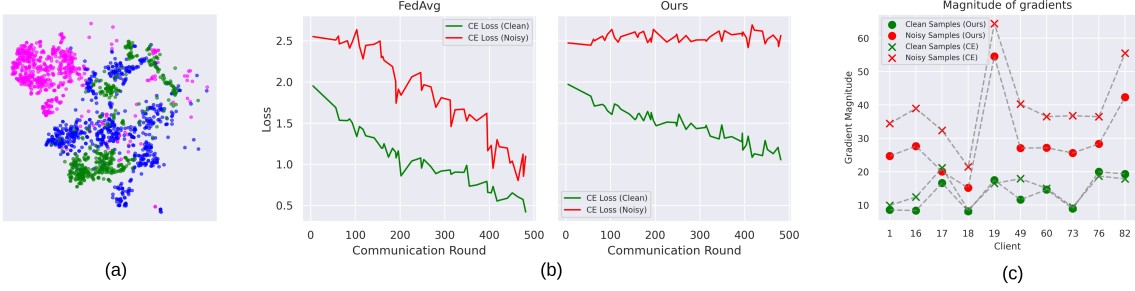

(a)          (b)          (c)

Figure 2: (a) t-SNE plot of features from the MURA dataset, extracted using an ImageNet SSL-pretrained model, showing three distinct classes, each represented by a different color, with visible sub-clusters. (b) Cross-Entropy loss curves for noisy and clean training samples of a randomly selected client in the (0.7, 0.5) noise setting trained with FedAvg and our method. (c) Magnitude of the gradient for clean and noisy samples across 10 high-noise clients in the (0.7, 0.5) noise setting .

---

**Algorithm 1** Federated Learning with Local $K$-Similarity Constraint Objective

---

1: INPUT: Server model $w = (g, h)$, local training epoch $E$, communication rounds $R$, learning rate $\eta$, total number of clients $N$, fraction of clients to select per round $F$, client $n$'s dataset $\hat{D}_n = \{(x_i, \hat{y}_i)_{i=1}^{M_n}\}$, SSL pretrained model $f$

2: **procedure** SERVERTRAINING( )

3:     initialize $w_0 = (g, h)$

4:     **for** each round $t = 1, 2, \ldots, R$ **do**

5:         $C \leftarrow \max(F \cdot N, 1)$

6:         $S_t \leftarrow$ (random set of $C$ clients)

7:         **for** client $n \in S_t$ **do**

8:             $w_t^n \leftarrow$ CLIENTTRAINING$(n, w_{t-1})$

9:         **end for**

10:        $w_t \leftarrow \sum_{n \in S_t} \frac{|\hat{D}_n|}{\sum_{i \in S_t} |\hat{D}_i|} w_t^n$

11:     **end for**

12: **end procedure**

13: **procedure** CLIENTTRAINING$(n, w)$                                             ▷ Run on client $n$

14:     B $\leftarrow$ (split $\hat{D}_n$ into set of batches of size $m$)

15:     **for** epoch $e = 1, 2, \ldots, E$ **do**

16:         **for** $\mathcal{B} = \{(x_{i_j}, \hat{y}_{i_j})_{j=1}^m\}$ in B **do**

17:             $w \leftarrow w - \eta * \frac{1}{m} \sum_{i=1}^m \left[ \nabla \mathcal{L}_{\text{CE}}(\hat{y}_{i_j}^n, h_n(g_n(x_{i_j}^n))) + \lambda * \nabla \mathcal{L}_{\text{CL},i_j}(X; f, g_n) \right]$

18:         **end for**

19:     **end for**

20:     return $w$ back to the server

21: **end procedure**

---

### 3.3   How local $K$-similarity constraint loss works?

In this section, we provide intuition on how the local $K$-similarity constraint objective enhances robustness to label noise. We can rewrite Equation 3 in terms of classifier space representations, $z^c$ as:

$$\mathcal{L}_{i_j}(x_{i_j}^n; X, \hat{Y}, f, g_n, h_n) = \mathcal{L}_{\text{CE}}(\hat{y}_{i_j}^n, h_n(z_{i_j}^c)) - \lambda \cdot \log \left( \frac{\sum_{i_k \in \mathcal{N}} \exp(\langle z_{i_j}^c \cdot z_{i_k}^c \rangle / \mathcal{T})}{\sum_{l=1}^m \exp(\langle z_{i_j}^c \cdot z_{i_l}^c \rangle / \mathcal{T})} \right). \tag{4}$$

The representation space of an SSL model exhibits fine-grained semantic sub-class clusters independent of label noise (Xue et al., 2022) (Figure 2 (a)). The term on the right of Equation 4 enforces this semantic structure on $z^c$, by utilizing the immediate neighbours $\mathcal{N}$ as positive pairs. As shown in Figure 3 (top), while the coss-entropy loss with noisy labels aims to disrupt semantic clusters (shown with red) (as also shown later in resulting dispersed representations of FedAvg in Figure 6), our constraint acts as a cohesive force (shown with green) bringing $z_{i_j}^c$ and semantically similar elements, $z_{i_k}^c$ together. Consequently, the learned representations are less biased towards noise, aiding generalization across noise-free data.

This regularization effect is illustrated in Figure 2 (b). In this figure, we compare the trend for cross entropy loss for clean and noisy samples with and without the use of our proposed regularization approach. To compute $L_{\text{CE}}(\text{noisy})$ and $L_{\text{CE}}(\text{clean})$, we first calculate per-sample contrastive loss across all training samples for each client. Using ground-truth noise annotations from our controlled setup, we divide the samples into clean and noisy groups. We then compute the average cross-entropy loss for each group independently. This analysis is based on the (0.7, 0.5) noise setting on CIFAR-10, where 70% of clients are noisy, and each contains at least 50% noisy samples.

Figure 2(b) shows that, unlike FedAvg, our structure-preserving loss prevents the cross-entropy loss on noisy samples from decreasing sharply. This stability indicates that the model avoids overfitting to label noise. Furthermore, we also analyzed the magnitude of gradients for both clea n and noisy labels for our proposed

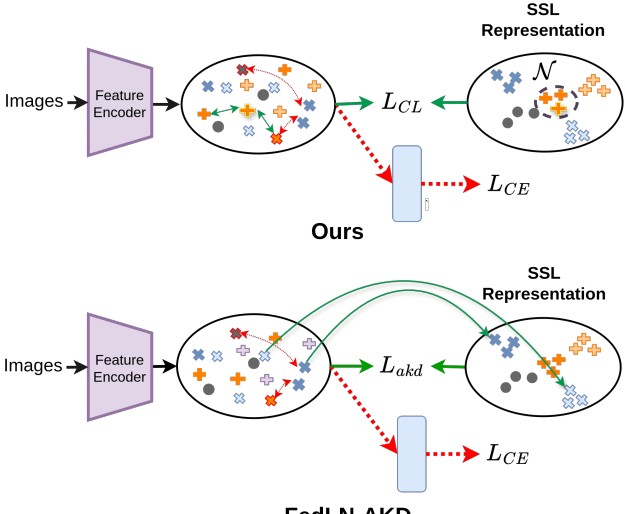

Figure 3: Comparison of our local regularization approach with FedLN-AKD visualizing 3 classes (blue, grey, and orange). Please zoom in for a better view. Colors represent ground truth labels while shapes with red borders represent samples with wrongly assigned labels-assigned labels represented by shape. Solid and hollow shapes of the same color refer to elements of different sub-class clusters belonging to the same parent class. Thin red arrows illustrate the tendency of CE to bring embeddings with the same assigned labels close. Thin green curves in FedLN-AKD illustrate the regression of embeddings according to the SSL space.

regularization compared to FedAvg. In order to learn from the correct labels and avoid memorization of noisy labels, it is necessary to reduce the contribution to the gradient from examples with noisy labels (Liu et al., 2020). As shown in Figure 2 (c), applying the local $K$-similarity constraint as a regularizer decreases the magnitude of the gradients for samples with noisy labels. Together, these findings demonstrate that our regularizer effectively mitigates the impact of label noise during training, both by preventing overfitting in the loss and reducing harmful gradients.

*Why FedLN-AKD is suboptimal?* FedLN-AKD (Tsouvalas et al., 2024), which is a directly comparable method to ours, utilizes L1 loss to directly enforce the embeddings to closely resemble SSL representations. However, the structure of SSL representations can result in sub-class clusters of the same parent class residing at different regions of the SSL space (as shown by solid and hollow sub-class clusters in Figure 3 and as observed in the SSL representation space of the MURA dataset in Figure 2 (a)). While this regularization is more robust compared to cross-entropy loss alone, the strict preservation of sub-class structures can restrict embeddings belonging to the same parent class from converging and increase the intra-class variance in the resulting representation space. In contrast, our method only enforces the immediate neighbourhood structure of the SSL representation space on the classifier representation space, avoiding such restriction. This offers a more robust approach to leverage SSL representations for classification tasks of diverse data domains, as demonstrated by the superior robustness of our method on domains beyond the SSL pretraining data.

## 4 Experiments

We conducted experiments on six publicly available image classification benchmarks: PathMNIST (Yang et al., 2023), MURA(Rajpurkar et al., 2017), CIFAR-10/100 (Krizhevsky et al., 2009) and their human annotated real-world noise versions, CIFAR-10N/100N (worst) (Wei et al., 2021), following standard train and test splits. While CIFAR-10(N) and CIFAR-100(N) contain images from the natural domain, MURA contains bone X-rays belonging to 7 different anatomical locations, and PathMNIST is a collection of H&E stained histological image patches categorized into 9 different tissue types. For our experiments, we employ a pretrained ResNet-50 model. This pretrained model was originally pretrained on ImageNet dataset (Deng

et al., 2009) using SimCLR (Chen et al., 2020) [1] . Additionally, we highlight the applicability of alternative SSL encoder architecture and method using a ViT-S/16 based image encoder pretrained using DINO (Caron et al., 2021). Unless explicitly mentioned otherwise, all studies with the "Ours" method utilize the SimCLR pretrained ResNet-50 model.

## 4.1 Implementation Details

| Dataset | CIFAR-10(N)/100(N) | PathMNIST/MURA |
|---|---|---|
| Dataset Size | 50000 | 89,996 / 36,808 |
| # of classes | 10 / 100 | 9 / 7 |
| Architecture | ResNet-18 / 34 | ResNet-18 |
| # of clients | 100 | 100 / 50 |
| Fraction | 0.1 | 0.1 |
| # Rounds | 1000 | 300 |
| Image Size | 32x32 | 128x128 / 224x224 |

Table 1: Implementation Details.

For the classifier models, we utilize randomly initialized ResNet-18 architecture for CIFAR-10(N), MURA and PathMNIST, whereas for CIFAR-100(N) we use ResNet-34. All experiments are performed using a local batch size of 50 and local training epochs set to 3 per round. SGD optimizer with a learning rate of 0.01 and weight decay of 3e-4 without learning rate decay and momentum was utilized for training the client models. For our approach, we set the temperature to 0.3, $K$ to 4, and $\lambda$ to 3 for all datasets and settings. All experiments were performed on NVIDIA A100 gpus. To ensure the reliability of results, all experiments were conducted with 3 random seeds. Besides MURA, where we report the best macro F1 score to capture performance on the imbalanced dataset, all other datasets are evaluated in terms of best accuracy. We refer to Table 1 for other experimental configurations.

| | $\rho$ | 0.7 | | 0.85 | | 1.0 | |
|---|---|---|---|---|---|---|---|
| | $\tau$ | 0.2 | 0.5 | 0.2 | 0.5 | 0.2 | 0.5 |
| CIFAR-10 (Krizhevsky et al., 2009) | FedAvg (McMahan et al., 2017) | 70.21±0.88 | 64.14±2.36 | 62.16±2.30 | 53.74±4.21 | 53.39±2.36 | 41.87±1.90 |
| | SymCE (Wang et al., 2019) | 79.70±0.76 | 73.00±1.64 | 74.07±2.90 | 62.40±5.24 | 65.36±2.58 | 48.22±2.42 |
| | LogitClip (Wei et al., 2023) | 73.12±1.28 | 66.65±1.91 | 66.91±3.11 | 58.49±3.85 | 60.62±2.31 | 47.10±1.54 |
| | FedNoro (Wu et al., 2023) | 79.84±2.13 | 77.42±2.36 | 72.31±5.00 | 69.12±2.41 | 60.84±2.51 | 47.49±3.0 |
| | FedCorr (Xu et al., 2022) | 80.43±1.27 | **79.97±2.08** | 73.04±2.70 | 71.24±2.86 | 64.55±0.47 | 47.64±7.23 |
| | FedLN-AKD (Tsouvalas et al., 2024) | **81.18±0.10** | 77.78±1.18 | 75.24±2.46 | 64.83±7.49 | 66.01±0.97 | 50.58±1.80 |
| | Ours | 80.96±1.05 | 79.31±1.43 | **77.68±1.68** | **73.91±3.99** | **73.92±1.45** | **64.91±0.44** |
| CIFAR-100 (Krizhevsky et al., 2009) | FedAvg (McMahan et al., 2017) | 37.36±1.96 | 31.81±2.14 | 31.28±2.59 | 24.63±4.07 | 25.79±1.41 | 15.86±1.11 |
| | SymCE (Wang et al., 2019) | 42.82±1.99 | 36.01±2.67 | 35.69±3.73 | 26.99±4.04 | 27.21±1.29 | 16.90±1.14 |
| | LogitClip (Wei et al., 2023) | 37.19±1.22 | 31.98±1.69 | 31.45±2.05 | 25.43±2.89 | 26.24±0.37 | 17.43±0.99 |
| | FedNoro (Wu et al., 2023) | 44.01±2.29 | 38.30±2.10 | 38.07±2.74 | 26.34±5.29 | 30.33±0.38 | 18.10±0.98 |
| | FedCorr (Xu et al., 2022) | 45.15±2.21 | 39.89±3.50 | 39.57±3.21 | 28.84±4.56 | 30.92±0.58 | 14.66±4.34 |
| | FedLN-AKD (Tsouvalas et al., 2024) | 50.94±2.73 | 42.36±4.38 | 36.20±8.17 | 23.33±7..34 | 28.59±1.91 | 15.26±0.66 |
| | Ours | **54.71±0.79** | **50.56±1.14** | **49.48±1.72** | **43.71±2.59** | **42.36±1.23** | **32.77±0.55** |
| PathMNIST (Yang et al., 2023) | FedAvg (McMahan et al., 2017) | 91.98±0.48 | 92.18±1.19 | 90.19±0.53 | 87.78±3.20 | 87.93±0.65 | 80.57±3.31 |
| | SymCE (Wang et al., 2019) | 93.95±0.65 | 93.55±0.29 | 93.66±0.12 | 92.59±0.79 | 93.39±0.42 | 90.26±0.46 |
| | LogitClip (Wei et al., 2023) | 91.88±0.08 | 91.24±0.83 | 90.85±0.04 | 88.24±1.47 | 88.78±1.29 | 83.15±2.28 |
| | FedNoro (Wu et al., 2023) | 91.58±1.55 | 90.73±1.83 | 89.47±0.82 | 87.00±1.31 | 89.96±0.93 | 85.03±1.52 |
| | FedCorr (Xu et al., 2022) | 93.19±1.28 | 93.06±0.56 | 92.63±0.96 | 91.77±2.89 | 91.15±0.79 | 86.59±3.00 |
| | FedLN-AKD (Tsouvalas et al., 2024) | **94.76±0.68** | 94.05±0.27 | 93.57±0.76 | 93.52±1.47 | 93.01±0.47 | 89.38±1.42 |
| | Ours | 94.61±0.21 | **94.20±0.21** | **94.44±0.22** | **93.74±0.64** | **93.58±0.26** | **91.94±0.77** |
| MURA (Rajpurkar et al., 2017) | FedAvg (McMahan et al., 2017) | 84.10±1.13 | 81.27±1.33 | 77.37±2.30 | 71.66±2.37 | 59.87±1.83 | 46.99±1.41 |
| | SymCE (Wang et al., 2019) | 87.38±0.87 | 84.75±1.38 | 82.84±1.93 | 77.78±0.67 | 65.59±1.60 | 45.95±7.21 |
| | LogitClip (Wei et al., 2023) | 83.18±1.25 | 80.28±2.05 | 75.60±2.15 | 70.05±1.39 | 64.82±3.59 | 52.07±0.94 |
| | FedNoro (Wu et al., 2023) | 88.33±0.07 | 86.84±0.70 | 84.07±1.69 | 79.17±8.40 | 70.89±1.97 | 54.08±2.43 |
| | FedCorr (Xu et al., 2022) | 87.44±0.76 | 86.86±2.63 | 82.82±2.97 | 83.01±1.87 | 70.89±2.22 | 54.76±1.98 |
| | FedLN-AKD (Tsouvalas et al., 2024) | 86.07±0.69 | 82.54±2.29 | 81.03±1.89 | 76.44±0.49 | 63.33±1.66 | 46.01±2.13 |
| | Ours | **89.88±0.49** | **88.34±0.89** | **85.67±1.40** | **83.22±2.34** | **78.09±0.87** | **71.91±0.79** |

Table 2: Best performance over 3 random seeds at non-IID configuration of $p = 0.7$, and $\alpha_{Dirchlet} = 5$. For MURA, scores are in macro F1, while for others, accuracy is evaluated.

**Data Distribution:** Following previous works (Xu et al., 2022), we utilize Bernoulli and Dirichlet distributions to generate non-IID data partitions. Specifically, we set $\alpha_{Dirichlet}$ to 5 and $\Phi_{ij} \sim$ Bernoulli(0.7). For

---

[1]Pretrained model obtained from `https://github.com/Lightning-Universe/lightning-bolts`

MURA, we perform split at the level of patients to prevent data leakage across clients. All reported results in the paper are in non-IID settings. We refer the reader to supplementary material for results in IID split, non-IID data, and noise distribution plots.

**Noise model:** Following previous works (Xu et al., 2022; Wu et al., 2023), we utilize two parameters to control the noise rate in the system: $\rho$ and $\tau$. Here, $\rho$ refers to the fraction of clients whose data is noisy and $\tau$ refers to the minimum noise rate in the noisy clients. Thus, each client has probability $\rho$ of being noisy, in which case the actual noise rate, $\mu_n$ is obtained by uniform sampling from $(\tau, 1)$ for each client $n$ independently, ensuring variation of noise rates across clients. We then assign a random label $\hat{y}$ among the $M$ classes in the dataset for $\mu_n\%$ samples of noisy clients chosen randomly. For all datasets, we perform experiments with $\rho \in \{0.7, 0.85, 1.0\}$. Accordingly, for all noisy client percentages $\rho$, we set the minimum noise level per client, $\tau \in \{0.2, 0.5\}$ for non-IID setting and $\tau = 0.5$ for IID setting. For simplicity, we refer to noise rates in the form $(\rho, \tau)$ in the results section. For CIFAR-10N/100N, we do not add any additional noise as these datasets already contain real-world noisy samples.

### 4.2 Baselines

We compare our approach with multiple prior works proposed in Federated Noisy Label Noise Learning: Fed-Noro (Wu et al., 2023), FedCorr (Xu et al., 2022), and FedLN-AKD (Tsouvalas et al., 2024). In addition, we uplift loss regularization approaches proposed in the Centralized Noisy Label Learning literature, LogitClip (Wei et al., 2023), and Symmetric Cross-Entropy Loss (SymCE) (Wang et al., 2019) to federated setting and provide comparisons. We ensure that the experimental configurations such as model architecture, number of local epochs, number of communication rounds, number of clients, and batch size were kept consistent across all baselines for a fair comparison.

## 5 Results

Table 2 reports the mean and standard deviation of the best accuracy obtained by the baselines and our model across multiple noise settings and datasets. We note that our approach generalizes to different data domains, as demonstrated by the superior performance in both computer vision and medical benchmark datasets. As observed, an increase in either $\rho$ or $\tau$ substantially reduces the performance of the existing methods. Notably, our approach is comparatively more robust to such deterioration, with the increase in $\rho$ and $\tau$. For instance, in CIFAR-10, the performance of our method deteriorates from 80.86 to 64.91 (16.05 points) as we change $(\rho, \tau)$ from (0.7, 0.2) to (1.0 0.5). Meanwhile, the performance of the second least deteriorating method for CIFAR-10, FedAvg, drops from 70.21 to 41.87 (28.34 points). Additionally, our method outperforms the baselines at nearly all noise settings and datasets. For instance, at $(\rho, \tau) =$ (1.0,0.5) our approach outperforms the second-best approach, FedLN-AKD by at least 14 points in CIFAR-10 and CIFAR-100 datasets. We obtain similar trends in both medical benchmark datasets, PathMNIST and MURA. Our approach brings over 2 points on improvement for PathMNIST, whereas, in MURA our method outperforms the existing approaches by over 17 points at the high noise setting.

### 5.1 Results on real-world noisy datasets

| Method | CIFAR-10N | CIFAR-100N |
|---|---|---|
| Fedavg | 64.37±0.86 | 43.16±0.66 |
| SymCE | 74.20±0.39 | 48.30 ±0.33 |
| LogitClip | 68.12±0.45 | 42.34±0.69 |
| FedNoro | 69.92±0.16 | 44.74±3.20 |
| FedCorr | 67.84±1.62 | 49.67 ±1.64 |
| FedLN-AKD | 73.79±0.21 | 50.00±0.83 |
| Ours | **74.48±0.16** | **51.73±0.26** |

Table 3: Best Accuracy on CIFAR-10N/100N datasets.

In Table 3, we report the best accuracy obtained by our approach and the baselines on CIFAR-10N and CIFAR-100N. These datasets contain real-world noisy samples that represent realistic *assymmetric* human-annotation error patterns. As observed, our approach provides significant performance gain over FedAvg and outperforms both centralized and federated noisy label learning baselines.

## 5.2 Study of hyperparameters

We present a study of the effect of $K$ on model performance in Figure 4 (left). We observe similar performance across small values of $K$ and set $K$ to 4 for all experiments. Increasing $K$ to larger values reduces the effectiveness of our method. We attribute this to the decrease in negative pairs and increase in semantically dissimilar positive examples in the contrastive objective as $K$ increases. Figure 4 (right) presents the effect of change in temperature, $\mathcal{T}$ on the best accuracy obtained by our approach. We observe $\mathcal{T} = 0.3$ to be optimal and use the same in all experiments.

In Figure 5, we study the effect of contrastive loss weight, $\lambda$ on the effectiveness of our approach. We also report the number of communication rounds required to achieve the best accuracy of 60%, to study the effect of $\lambda$ on model convergence. We observe that the regularization by our similarity constraint causes the model convergence to slow down as $\lambda$ is increased. We obtain best results at $\lambda = 3$ considering both convergence and performance.

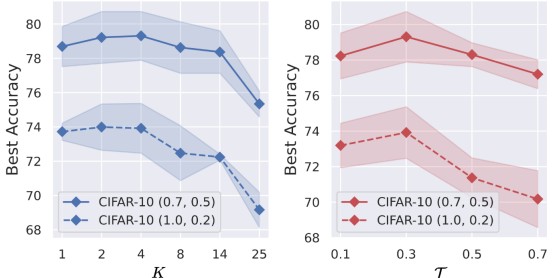

Figure 4: Best Accuracy on CIFAR-10 at various values of $K$ at $\mathcal{T}$ set to 0.3 (left) and various values of $\mathcal{T}$ at $K$ set to 4 (right).

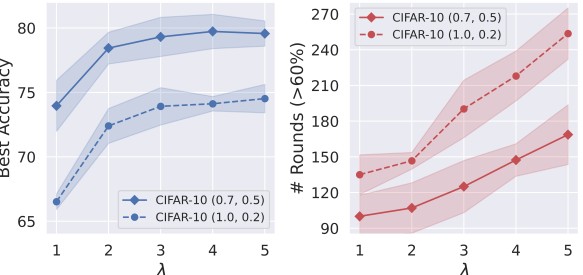

Figure 5: Best Accuracy and convergence rate reported at various values of $\lambda$ for $k = 5$ for CIFAR-10.

## 5.3 Ablation study

In Table 4, we compare results from using the SSL pretrained model with regularization caused by the constraint without a proper reference embedding space. Specifically, we apply the contrastive objective with respect to the representation space of a randomly initialized model. As observed, enforcing the constraint on the representation space of an untrained model does not always provide a definite benefit. Meanwhile, enforcing the constraint with respect to the representation space of the SSL pretrained model brings consistent improvements over FedAvg for both datasets.

| Method | CIFAR-10 (0.7, 0.5) | PathMNIST (1.0, 0.2) |
|---|---|---|
| Ours (w SimCLR model) | **79.31±1.43** | **93.58±0.26** |
| Ours (w random model) | 71.67±1.76 | 87.86±0.50 |
| FedAvg | 64.14±2.36 | 87.93±0.65 |

Table 4: Ablation study of reference regularization space.

### 5.4   SSL Pretrained Initialization

Past works in federated noisy label learning have shown the effectiveness of using SSL pretrained models to initialize the classifier model. We present a study to validate the effectiveness of our approach when compared to initialization from the SSL pretrained model in Table 5. For this study, we set the client architecture to be the same as that of the SimCLR pretrained model.

While initialization using a pretrained model(row 2) performs well against label noise, we consider the requirement of client architecture to be the same as that of pretrained models to be limiting. Notably, our approach performs competitively compared to pretrained initialization, even surpassing pretrained initialization for PathMNIST. Additionally, our method is orthogonal to pretrained initialization and performs well when applied in addition to the pretrained initialization approach, as seen in Table 5.

| Method | CIFAR-10 (0.7, 0.5) | PathMNIST (1.0, 0.2) |
|---|---|---|
| FedAvg | 60.37±0.60 | 89.32±0.83 |
| FedAvg + Pretrained Init | 75.35±1.22 | 90.67±0.15 |
| Ours | 72.83±0.76 | 92.99±0.49 |
| Ours + Pretrained Init | 77.70±0.43 | 96.13±0.46 |

Table 5: Pretrained initialization study with ResNet-50 model pretrained with SimCLR, client architecture is also set to ResNet-50.

### 5.5   Encoder architecture agnostic

The use of separate encoder architectures for the SSL pretrained model (ResNet-50) and classifier models (ResNet-18 and ResNet-34) shows that our framework allows for different architectures between the pretrained encoder and client models. To further verify the model-agnostic nature of our framework and highlight the applicability of other SSL encoders, we present a study replacing the SimCLR pretrained ResNet-50 model with a ViT-S/16 model pretrained using DINO in Table 6. As observed, our method works well with the DINO encoder. Notably, we note a sharp increase in the performance of the model utilizing DINO possibly due to better pretrained features.

| Method (Ours) | CIFAR-10 (0.7, 0.5) | PathMNIST (1.0, 0.2) |
|---|---|---|
| w SimCLR(ResNet-50) | 79.31±1.43 | 93.58±0.26 |
| w DINO(ViT-S/16) | 86.34±0.13 | 94.02±0.16 |

Table 6: Study with pretrained ViT-S/16 DINO encoder.

### 5.6   Qualitative Analysis

In Figure 6, we present t-SNE plots of the representation spaces of the global model after training with FedAvg and with our method utilizing the local $K$-similarity constraint. While the representation space produced by FedAvg and FedLN-AKD appear highly dispersed, training with our constraint at high noise settings produces more compact and fine-grained clusters. In Figure 7, we present samples of positive and negative examples obtained by using the batch-wise $K$-nearest neighbourhood $\mathcal{N}$. As observed the anchor and positives tend to be visually and semantically similar while the negatives tend to be different.

## 6   Conclusion

In this work, we presented an approach to locally regularize client models by leveraging the feature representations of self-supervised pretrained models in a federated setting with label noise. Experiments across various noise levels and datasets demonstrate the effectiveness of our approach compared to existing methods.

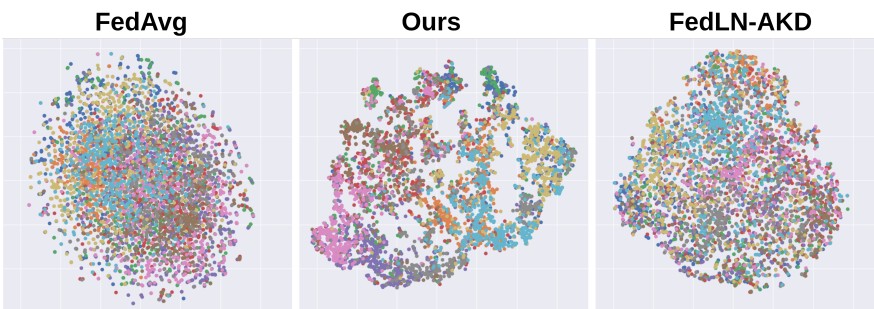

Figure 6: t-SNE plots of representation space obtained by the final global model trained on FedAvg (left), FedLN-AKD (right) and with our local $K$-similarity constraint (Center) at (1.0, 0.5) noise configuration on CIFAR-10. Please zoom in for better view.

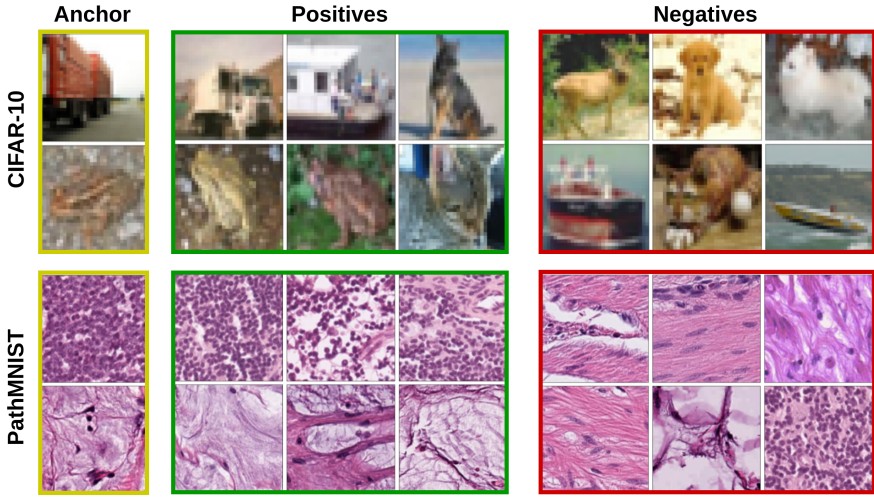

Figure 7: Positive and negative samples obtained by using $K$-nearest neighbourhood, $\mathcal{N}$ at (1.0, 0.2) noise configuration on CIFAR-10 and PathMNIST.

Notably, our approach generalizes well to both natural and medical image classification benchmarks. As our method uses the SSL model solely for regularization, both the client model and SSL pretrained model can be independently selected, making it architecture-agnostic. However, the effectiveness of this regularization depends on the quality of the SSL model's representations. Future research could explore other SSL models pretrained on diverse datasets to further assess their impact on the regularization of federated clients.

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

# A Appendix

## A.1 Outline

As part of the supplementary material for "Local $K$-Similarity Constraint for Federated Learning with Label Noise", we provide additional details and experimental results as organized below:

- Appendix A.2 analyzes the dependency of our approach on pre-trained SSL models.
- Appendix A.3 discusses additional overhead introduced by our method.
- Appendix A.4 provides additional implementation details, particularly regarding the various baselines used in the paper.
- Appendix A.5 provides results in IID setting for CIFAR-10N and CIFAR-100N benchmarks.
- Appendix A.6 provides comparison of our approach with the baselines in IID setting for CIFAR-10, CIFAR-100, PathMNIST and MURA datasets.
- Appendix A.7 presents t-SNE plots of the reference representation space (of SimCLR pretrained model) for CIFAR-10(N), CIFAR-100(N), PathMNIST and MURA.
- Appendix A.8 presents the plots for non-IID data and noise distributions for CIFAR-10, CIFAR-100 and PathMNIST datasets.
- Appendix A.9 provides additional samples of positive and negative examples retrieved from the representation space of SimCLR pretrained ResNet-50 model.

## A.2 Dependency on Quality of pre-trained SSL models

The performance of our framework is naturally influenced by the quality and domain-relevance of the pre-trained SSL model's representations. However, our contribution is not in selecting the optimal SSL model but in proposing a flexible and effective strategy to leverage readily available pretrained encoders for robust federated learning under label noise. This flexibility is evidenced by our experimental results: even when using SSL models trained on natural images, we observe substantial robustness improvements on medical datasets (Table 2), outperforming baselines across all noise settings. Moreover, our method is architecture-agnostic, i.e., any pretrained encoder that outputs representations can be seamlessly incorporated in place of those used in our experiments, we we only require the representations from these models and not actual parameters. This can turn reliance on SSL models into a strength. For instance, substituting SimCLR with DINO yields substantial robustness improvement across both natural and medical domains (Table 6). These results highlight that advances in SSL pretraining can directly translate into improved robustness within our framework. We acknowledge that domain mismatch may affect the degree of improvement. Based on Table 2, we present a plot comparing percentage gain in performance across different datasets at various noise rates (Figure 8 Specificaly, we calculate

$$\frac{\text{Mean Performance(Ours)} - \text{Mean Performance(FedAvg)}}{\text{Mean Performance(FedAvg)}} * 100\%$$

at each noise setting and present a comparison across datasets. We observe that in-domain (natural) SSL models provide larger gains on natural datasets (in domain) than on medical datasets (cross domain). This suggests that SSL models pretrained on large-scale medical data, once more widely available, could further amplify performance for medical datasets. Nevertheless our experiments suggest that even under mismatched conditions, our approach consistently improves robustness. Furthermore, even using a randomly initialized model in place of the SSL model(Table 4) does not substantially harm model robustness, as suggested by comparable performance in PathMNIST to FedAvg, and substantial gain for CIFAR10. Thus, while the framework does depend on SSL representations, its stability and effectiveness hold across domains, and its relevance will only grow as better domain-specific SSL models become increasingly accessible.

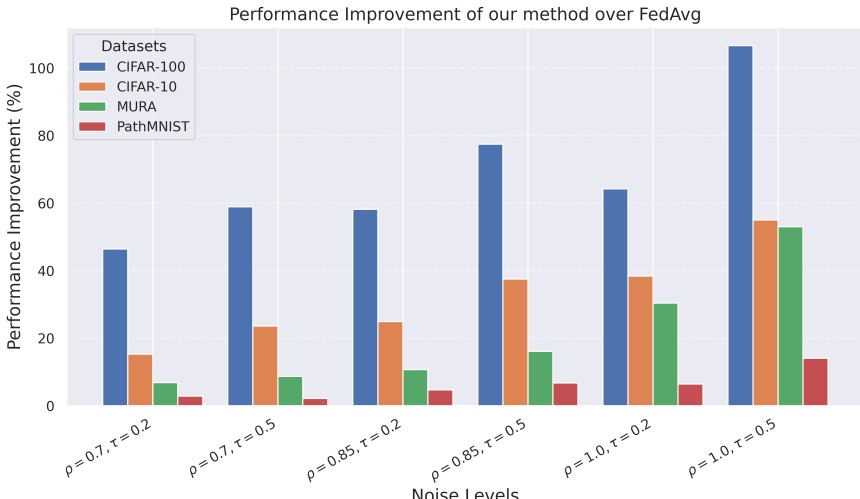

Figure 8: Percentage improvement in robustness of our method over FedAvg across different noise levels and datasets.

## A.3 Computation Overhead

The additional computational cost of our method primarily stems from two components: (i) extracting representations of input samples using a pre-trained SSL model, and (ii) computing the contrastive regularization objective. While our regularization objective does introduce additional overhead compared to vanilla FedAvg during the local training, we would like to highlight that no overhead is placed during inference or on the communication cost, while provide substantial robustness improvement. We report the per-epoch runtime in Table 7 against FedAvg the simplest federated learning algorithm with no mechanism for loss handling. to quantify this overhead.

| Method | Running Time (sec) |
|---|---|
| FedAvg | 8.94 |
| Ours (w SimCLR) | 21.35 |

Table 7: Per-epoch runtime (in seconds) comparison between our method and FedAvg on the CIFAR-10 dataset using a ResNet-18 backbone. All experiments were conducted on a single NVIDIA A100 GPU.

## A.4 Baseline Implementation Details

For all baselines, unless stated otherwise, we utilize the same settings of local batch size of 50, SGD optimizer with a learning rate of 0.01, and weight decay of 3e-4 without momentum. For CIFAR-10(N)/100(N) we set the effective communication round to 1000, while for PathMNIST and MURA, the effective number of communication rounds is set to 300 for all baselines. For fair comparison, we utilize the same augmentation settings across all methods. We mention other method-specific configurations below:

- **SymCE**: For all datasets, we set the weights of both cross entropy and reverse cross-entropy ($\alpha$ & $\beta$) to 0.5.
- **LogitClip**: For all settings, we set the upper bound of the norm of the logit vector ($\tau$) to 1.0.
- **FedCoRR**: For CIFAR-10(N)/100(N), we set $T1$ to 5 iterations, $T2$ to 500 communication rounds, and $T3$ to 450. Similarly for PathMNIST, we set $T1$ to 5 iterations, $T2$ to 125 rounds,z, and $T3$ to 125 rounds.
- **FedNoro**: For all datasets, we set the warmup rounds ($T_1$) to 50.Other hyperparameters were configured as mentioned in the original paper.

- **FedLN-AKD**: For all settings, the learning rate is set to 0.1. We set the distillation loss weight, $\alpha_{akd}$ to 10 for lower noise levels of $\rho \in \{0.7, 0.85\}$, whereas for higher noise level of $\rho = 1.0$, we set $\alpha_{akd}$ to 3.

## A.5 Results on CIFAR10N/100N in IID setting

| Method | Dataset | |
|---|---|---|
| | CIFAR-10N | CIFAR-100N |
| Fedavg | 66.153±0.81 | 42.73±0.3 |
| SymCE | 75.33±0.09 | 48.3 ±0.45 |
| LogitClip | 71.07±0.31 | 41.72±0.74 |
| FedNoro | 69.87±0.87 | 46.83±0.28 |
| FedCorr | 75.41±0.16 | 50.58 ±0.58 |
| FedLN-AKD | 75.48±0.61 | 51.50±0.63 |
| Ours | **76.53±0.38** | **52.28±0.14** |

Table 8: Best Accuracy on CIFAR-10N and CIFAR-100N datasets in IID setting.

In the main paper, we have provided the comparisons of our approach with baselines for real-word noisy datasets, CIFAR-10N and CIFAR-100N. To further validate the effectiveness of our approach, we present comparisons in the IID setting in Table 8. As observed, our approach outperforms the different baselines in the IID setting as well.

| | $(\rho, \tau)$ | (0.7, 0.5) | (1.0, 0.5) |
|---|---|---|---|
| | FedAvg | 68.26±2.17 | 45.01±1.37 |
| | SymCE | 76.52±2.50 | 50.02±1.78 |
| | LogitClip | 70.52±2.11 | 49.51±1.44 |
| CIFAR-10 | FedNoro | 79.51±1.72 | 51.43±1.28 |
| | FedCoRR | 80.55±2.26 | 48.68±1.42 |
| | FedLN-AKD | 80.70±0.48 | 51.80±2.32 |
| | Ours | **81.44±0.54** | **66.76±0.52** |
| | FedAvg | 32.26±1.57 | 15.63±0.81 |
| | SymCE | 34.53±2.95 | 17.72±1.00 |
| | LogitClip | 31.09±0.92 | 17.88±0.93 |
| CIFAR-100 | FedNoro | 38.28±1.08 | 21.15±1.36 |
| | FedCoRR | 40.35±2.90 | 18.26±0.78 |
| | FedLN-AKD | 38.28±2.10 | 9.99±1.18 |
| | Ours | **51.40±1.15** | **33.27±1.69** |
| | FedAvg | 91.27±0.98 | 84.11±1.80 |
| | SymCE | 93.51±0.78 | 90.85±0.29 |
| | LogitClip | 91.02±1.44 | 85.89±1.01 |
| PathMNIST | FedNoro | 91.82±0.86 | 86.44±0.39 |
| | FedCoRR | 94.20±0.50 | 88.74±0.88 |
| | FedLN-AKD | 94.38±0.33 | 91.20±0.81 |
| | Ours | **94.80±0.16** | **94.06±0.51** |
| | FedAvg | 80.13±2.73 | 51.18±1.98 |
| | SymCE | 85.00±0.77 | 53.66±3.42 |
| | LogitClip | 79.98±0.75 | 54.15±1.38 |
| MURA | FedNoro | 87.60±1.52 | 59.30±0.96 |
| | FedCoRR | **89.12±0.72** | 59.48±1.32 |
| | FedLN-AKD | 83.47±1.49 | 49.19±2.71 |
| | Ours | 88.53±0.36 | **73.45±1.19** |

Table 9: Best performance reported at different noise levels with IID setting over 3 different seeds. Macro F1 is reported for MURA and accuracy is reported for other datasets.

### A.6 Results on synthetic noise benchmarks in IID setting

We additionally present comparisons on the IID setting for the different datasets. We report the results at noise settings of $\rho \in \{0.7, 1.0\}$ and $\tau = 0.5$ in Table 9. Results show that our approach also outperforms the baselines at the different noise distributions for IID data distribution.

### A.7 Representation space of SimCLR pretrained ResNet-50 model

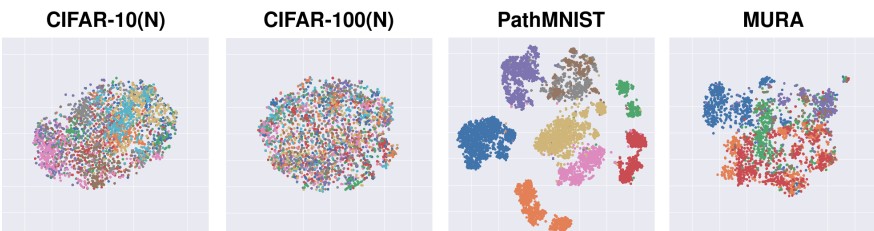

Figure 9: t-SNE plots of representation space (in test set) obtained from the SimCLR pretrained ResNet-50.

In Figure 9, we present the t-SNE plots of reference representation space for the different datasets.

### A.8 Data and Noise Distribution

In Figure 10, we present the class distribution across clients for non-IID splits. In Figure 11, we present the noise pattern of the CIFAR-10N and CIFAR-100N benchmarks, while in Figure 12, we present the noise distribution for 5 randomly selected clients on synthetic noise benchmarks.

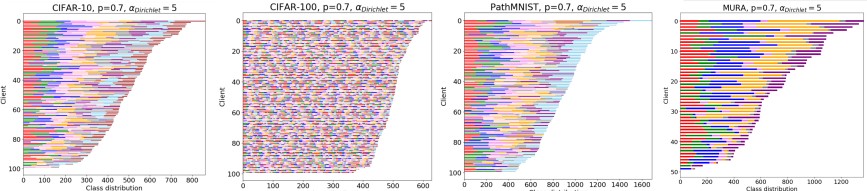

Figure 10: Data distribution across clients. Different colors represent different classes.

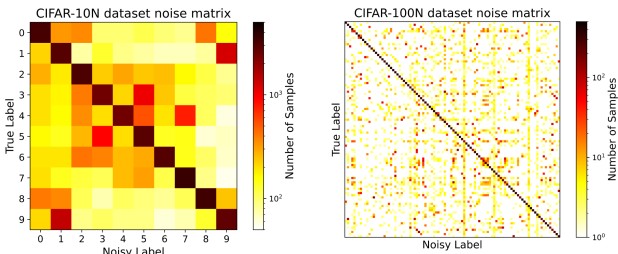

Figure 11: Noise Distribution of whole dataset for CIFAR-10N and CIFAR-100N

### A.9 Additional samples of retrieval

Figures 13 and 14 present additional samples of positive and negative examples retrieved by our approach from the reference representation space of the SSL pretrained model.

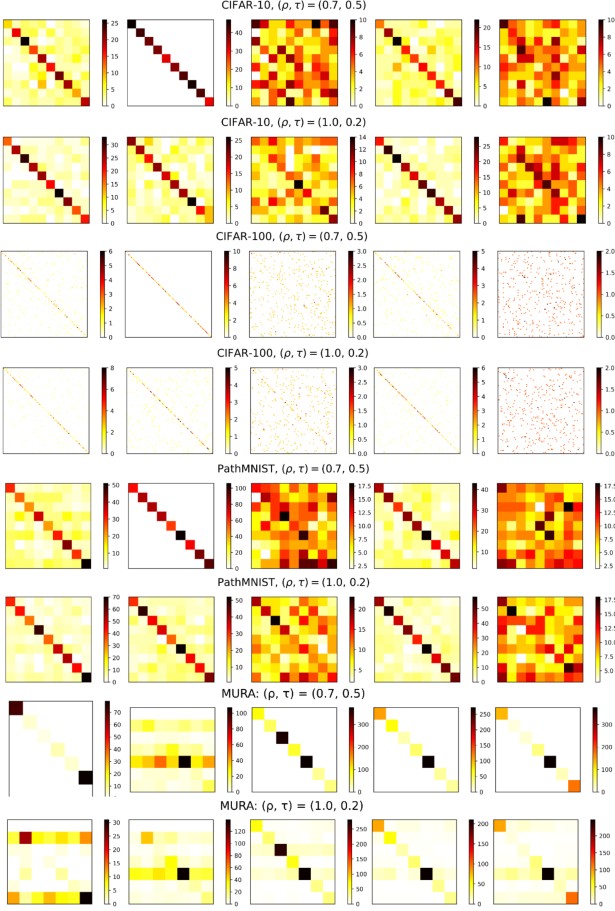

Figure 12: Noise distribution for 5 sample clients at various noise settings.

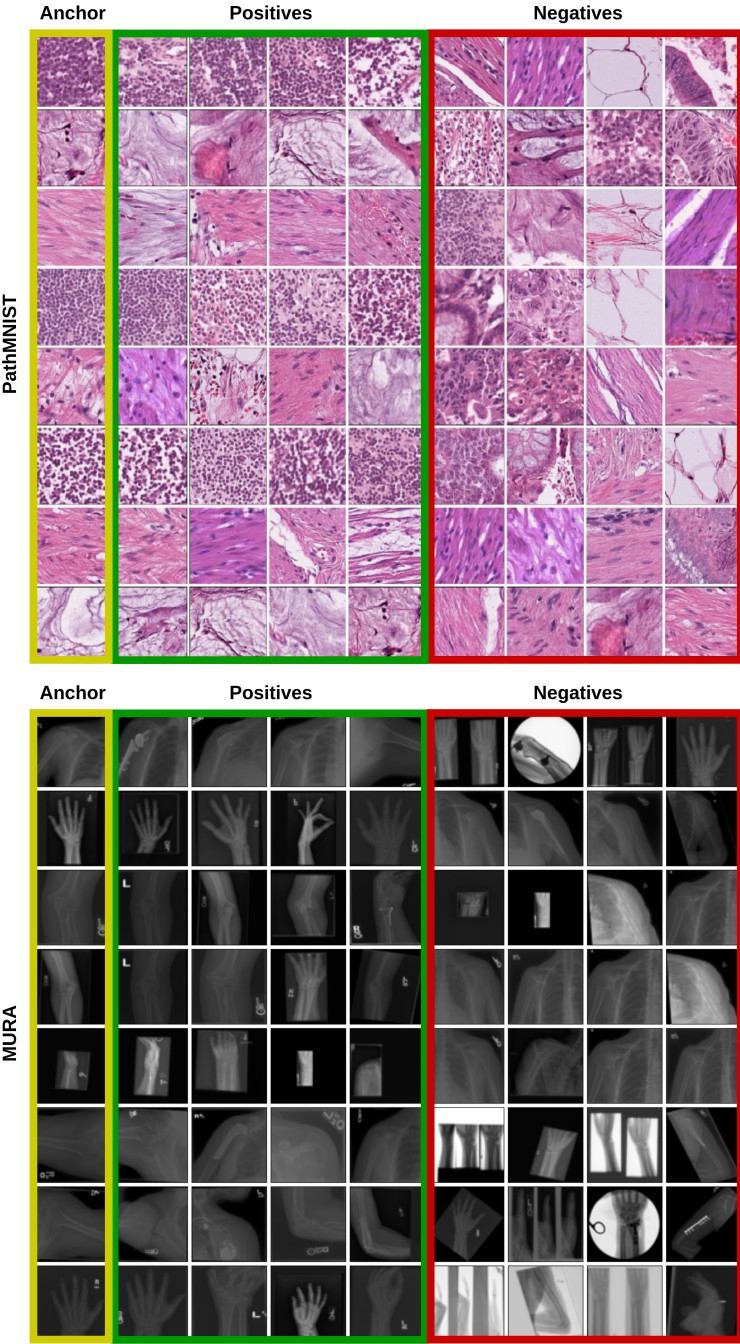

Figure 13: Additional examples of batch-wise positives and negatives retrieved from SimCLR pretrained ResNet-50 for PathMNIST and MURA.

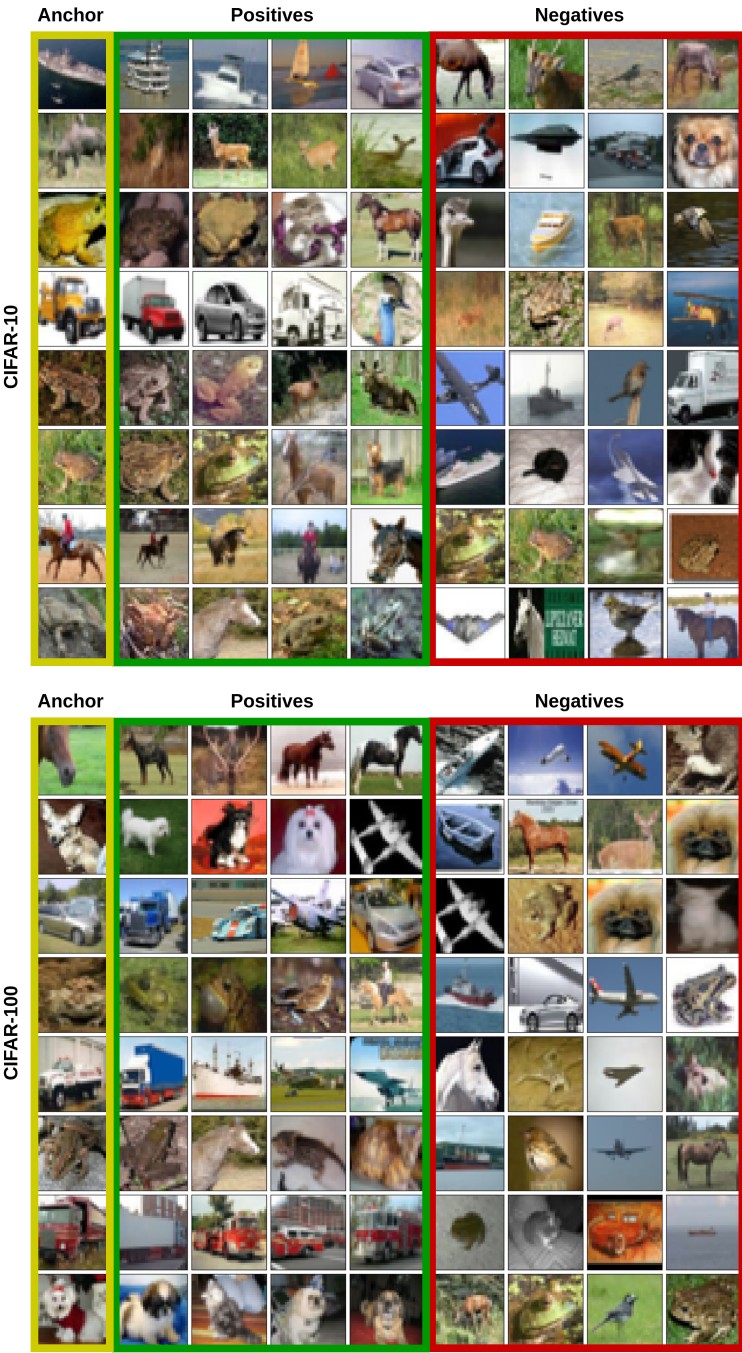

Figure 14: Additional examples of batch-wise positives and negatives retrieved from SimCLR pretrained ResNet-50 for CIFAR-10 and CIFAR-100.

