# OpenReview forum: "Local K-Similarity Constraint for Federated Learning with Label Noise"
_TMLR — Rejected by TMLR_

### Review · Reviewer_TU4L · 2025-08-06

**Summary Of Contributions:**

This paper proposes augmenting the loss function of a federated learning (FL) framework with an additional contrastive loss term L_{CL}, aiming to mitigate the impact of noisy data on performance. The authors conduct extensive experiments to demonstrate the effectiveness of the proposed approach.
However, there are still some concerns regarding the proposed approach. The primary challenge lies in the reliance on the self-supervised learning (SSL) model. The overall performance of the framework is highly dependent on the effectiveness of this component.
When the SSL model provides well-separated embeddings, the combined loss function (i.e.,L_{CE} + L_{CL}) is likely to be more effective. What happens when the SSL model is pretrained on a dataset different from the target dataset used by the classifier? Will the framework still be effective?
Additionally, employing an SSL model on each client introduces extra computational overhead, which should also be addressed and discussed in the paper.

**Audience:**

Yes

**Audience Explanation:**

The paper addresses a challenge in federated learning: how to improve model robustness in the presence of noisy or corrupted client data using contrastive learning. Given the growing interest in trustworthy and efficient federated learning systems, this topic is likely to appeal to a subset of TMLR's readership.

**Claims And Evidence:**

Yes

**Claims Explanation:**

Partially.
The paper presents an approach by incorporating a contrastive loss term L_{CL} into the federated learning objective to mitigate the impact of noisy data. The authors conduct extensive empirical experiments under various settings (e.g., IID, non-IID, different levels of label noise) to demonstrate the effectiveness of their method. The results show consistent improvements over several baselines, which supports their main claim.
However, there are still some concerns regarding the proposed approach. The primary challenge lies in the reliance on the self-supervised learning (SSL) model. The overall performance of the framework is highly dependent on the effectiveness of this component. Additionally, employing an SSL model on each client introduces extra computational overhead, which should also be addressed and discussed in the paper.

**Requested Changes:**

The primary challenge lies in the reliance on the self-supervised learning (SSL) model. The overall performance of the framework is highly dependent on the effectiveness of this component.
What happens when the SSL model is pretrained on a dataset different from the target dataset used by the classifier? Will the framework still be effective? This potential limitation should be addressed clearly in the paper. Please discuss the assumptions needed for the success of the framework, especially regarding the relationship between the SSL pretraining data and the downstream classification task.
 While I understand that the paper is more application-oriented and does not require deep theoretical analysis, some justification for convergence should be included. In particular, does addingL_{CL} from a mismatched SSL model risk destabilizing training or leading to divergence?

Other Comments and Questions:
1. How many samples does each client have in the IID setting for different cases?
2. In the non-IID setup, do all clients have the same number of samples? If not, consider showing a histogram of client sample sizes to help understand the data distribution.
3. The algorithm currently placed in the appendix. It is better to move it into the main paper.
4. In the algorithm pseudocode (in line 17), it should add “+” between L_CE and L_CL.
5. Why is the batch size set to 50? Please explain why this value was chosen over more conventional sizes like 32 or 64.
6. When comparing your method against others in Table 2, are all experimental settings consistent? For example, do you use the same number of clients, batch size, model architecture, and number of local epochs?
7. During local training, how are positive and negative samples constructed for each sample? It appears that for each sample in the experiments, there are k=4 positive and 50−4 negative samples. This results in a large imbalance between negative and positive samples when computing L_{CL} in training.
Why not use an equal number of positive and negative samples, or alternatively, use a weighted version of L_{CL} to balance their contributions? A rationale is needed for this design choice.
8. In Figure 2b, how is L_{CL} computed separately for noisy and clean samples within a client?
For example, does L_{CL}(noisy) represent the sum over all noisy samples' contrastive losses? What is the number of total samples, noisy samples, and positive samples per client in this figure? Please provide more detailed explanation on this figure.

---

> ### Author Response · Authors · 2025-09-01
> **Response to Reviewer TU4L**
>
> Thank you for your thoughtful comments and constructive feedback. Below we address key concerns:
>
> ### Requested Changes
> >…reliance on SSL model …relationship between SSL pretraining data and downstream classification task …some justification for convergence
>
> Thank you for your suggestions. Appendix A.2 now discusses our method's dependency on pretrained SSL model. We agree that a convergence analysis would provide additional insights, and we consider this a part of future work. However, our method's strong empirical results across a wide range of datasets and noise setups strongly suggests that it is effective and stable.
>
> We do not think that a mismatched SSL model can harm robustness significantly. Our experiments in Table 4(also placed below) showed that even with a randomly intialized ResNet50, the training converges, and the resulting accuracy is not significantly worse than FedAvg. Table 2 further demonstrates strong cross-domain generalization, where an SSL model trained on natural images significantly improves label-noise robustness for medical data.
> |Method|CIFAR10(0.7,0.5)|PathMNIST(1.0,0.2)|
> |-|-|-|
> |Ours(w SimCLR model)|79.31±1.43|93.58±0.26|
> |Ours(w random model)|71.67±1.76|87.86±0.50|
> |FedAvg|64.14±2.36|87.93±0.65|
>
> Table 4:Study of reference regularization
>
> ### Computational Overhead
> The extra cost of our method comes from (i)representation extraction with pretrained SSL model and (ii)computing contrastive objective. Table below compares runtime against FedAvg, the simplest federated algorithm without loss-handling mechanism. While our method introduces some overhead during local training, it adds no cost at inference or in communication, yet provides substantial robustness improvements. We have also added this analysis to Appendix A.3.
> |Method|Per epoch runtime(sec)|
> |-|-|
> |FedAvg|8.94|
> |Ours(w SimCLR)|21.35|
>
> Table:Per epoch runtime on CIFAR10 with ResNet18.
> ## Other Comments and questions
> ### 1. No. of samples in IID
> In all IID setups, each client has uniform no. of samples. For CIFAR10, 50000 samples are split evenly across 100 clients, giving 500 per client.
>
> ### 2. No. of samples in non-IID
> In non-IID setup, clients have varying no. of samples and class distributions. We refer the reviewer to Figure 9 in the Appendix.
>
> ### 3. Moving algorithm
> We have moved the algorithm to main paper.
>
> ### 4. Consistency in setups
> Yes, all experimental settings are consistent across all baselines and noise setups in the tables. We have added this clarification in Sec. 4.2.
>
> ### 5. '+' in algorithm
> Thank you. We have made a correction.
>
> ### 6. Reasoning of batch size=50
> Our choice was based on our infrastructural setup and codebase of the paper we used, which also used batch size=50 for non-IID CIFAR10 experiments[1].
>
> [1] Tang, Minxue, et al. "FedCor: Correlation-based active client selection strategy for heterogeneous federated learning." CVPR 2022.
>
> ### 7. Selection of positive and negative samples, imbalance between no. of negatives and positives, alternatively why not use a weighted version of $L_{CL}$
> As we mention in 3rd paragraph of methodology(Sec. 3.2), we identify $k=4$ nearest neighbours in the SSL representation space as positive samples and all other batch elements as negative samples.
>
> We selected $k$ based on a hyperparameter study, shown in Fig. 4 (also placed below). Our study shows that large $k$ can degrade performance. We hypothesize in Sec. 5.2 that this degradation is likely due to increase in dissimilar positives in the contrastive objective. Furthermore, constrastive learning typically use large no. of negative samples than positive samples to capture finer details and subtle distinctions.
>
> Our implementation follows the widely used InfoNCE objective, which doesn't weight positive and negative samples separately. In fact, our objective reduces to InfoNCE when $k=1$. While a weighted strategy could be interesting, we leave this for future exploration.
>
> |k|Best Accuracy(%)|
> |-|-|
> |1|78.70±1.18|
> |2|79.22±1.51|
> |**4**|**79.31±1.43**|
> |8|78.63±1.50|
> |14|78.38±1.25|
> |24|75.34±0.77|
>
> Table: Study of k for CIFAR10(0.7,0.5)
>
> ### 8. Computing $L_{CL}$ separately for noisy and clean samples in Fig 2b. More details on this fig.
> We clarify that we plot the cross entropy(CE) loss, $L_{\text{CE}}$ in Fig. 2b. To compute $L_{\text{CE}}(\text{clean})$, and $L_{\text{CE}}(\text{noisy})$,  we first compute the per-sample CE loss for all training samples of a client. We then group the samples into clean and noisy subsets (using known noise labels used in our controlled experiments), and compute the mean CE loss separately. Specifically, $L_{\text{CE}}(\text{noisy})$ represents the average CE loss across all noisy samples in a client, and $L_{\text{CE}}(\text{clean})$ is for the clean samples. This study was performed in the (0.7,0.5) noise setting in CIFAR10, where 70% of the clients are noisy ≥ 50% noisy samples. We thank the reviewer and have revised the explanation in Sec. 3.3.

---

### Review · Reviewer_2uGV · 2025-08-06

**Summary Of Contributions:**

This paper deals with federated learning (FL) in the presence of label noise. Dealing with label noise is a highly active area of research, both in FL and in centralized learning (CL). As the main contribution of the paper, the authors proposed a simple regularization term, based on the idea of minimizing the contrastive loss relative to pairs of feature vectors from a separate pre-trained self-supervised model shared by all clients. Specifically, as part of the contrastive learning set-up, the authors used $K$ nearest neighbors in the feature space for positive pairs, where $K$ is a hyperparameter.

In the experiments section (Sec. 4), the authors have explicitly stated that an existing code implementation of SimCLR (Chen et al. 2020) was used for their proposed contrastive regularization term. In particular, the contrastive loss objective used is the same as the contrastive loss objective from SimCLR.

Overall, the proposed methodology can be effectively summarized as a straightforward application of contrastive learning to FL, via a simple contrastive regularization term, for enhancing the robustness to client label noise.

**Audience:**

Yes

**Audience Explanation:**

As explained in the summary of contributions, dealing with label noise is a highly active area of research, both in federated learning (FL) and in centralized learning (CL). I appreciate the authors' effort in carrying extensive experiments across 6 datasets to empirically verify that incorporating contrastive learning into FL does indeed improve classification accuracy. Probably the most helpful finding of this paper, at least to some TMLR readers, is an explicit quantification of the significant improvement in classification accuracy, in the extreme case when *all* clients have at least $50\\%$ label noise rate.

**Broader Impact Concerns:**

I have no concerns on the ethical implications of this work.

**Claims And Evidence:**

Yes

**Claims Explanation:**

I am indicating my answer as "yes", because the majority of the claims made in the paper are accurate, convincing, and clear. The authors have carried out extensive experiments on 6 datasets, across sufficiently diverse heterogeneous label noise settings, and compared their proposed method with 6 baselines. Sensitivity analyses for the hyperparameter $K$ and the temperature $\\mathcal{T}$ (another hyperparameter for the contrastive loss regularization term) are provided. An ablation study is also given, to better understand the contribution of the proposed contrastive regularization term. Overall, the experiment results do comprehensively justify the main claim that incorporating contrastive learning into FL would improve performance.

However, there are multiple inaccuracies that should be corrected; see requested changes below.

**Requested Changes:**

**1) Multiple mathematical/technical inaccuracies.** There are numerous (fortunately minor) mathematical/technical inaccuracies in this paper that should be corrected, before I can recommend acceptance.

- page 3, last 2 lines: "... where each $n$-th client has its separate training dataset $D_n = \\{(x_i^n, y_i^n)\\}_{i=1}^{M_n}$, such that each $i$-th data instance is a tuple consisting of input $x_i^n$ and ground truth label $y_i^n$."
- page 4, 2nd line after eqn (1): "$\\hat{D} \_n = \\{(x_i^n, \\hat{y} \_i^n)\\}_{i=1}^{M_n}$"
- page 4, Sec. 3.2, 2nd paragraph, line 2: "... comprising a feature extractor $g_n$ and a task-specific head $h_n$ (e.g., a linear classifier) for each client $n$."
- page 4, Sec. 3.2, 2nd paragraph, line 3: "... in a noisy client $n$, a minibatch $\\mathcal{B} = \\{ (x \_{i\_j}^n, \\hat{y} \_{i\_j}^n) \\}_{j=1}^m$ contains ..."
- page 4, Sec. 3.2, 2nd paragraph, line 4: "$X = \\{x \_{i_1}^n, x \_{i_2}^n, \\dots, x \_{i_m}^n\\}$ and their noisy labels $\\hat{Y} = \\{\hat{y} \_{i_1}^n, \\hat{y} \_{i_2}^n, \\dots, \\hat{y} \_{i_m}^n\\}$."
- page 4, Sec. 3.2, 2nd paragraph, line 5: "... both the feature encoder and the feature extractor (i.e., $f$ and $g_n$) ..."
- page 4, Sec. 3.2, 2nd paragraph, lines 5--7: "... two embedding sets: $f(X) := \\{z_{i_1}^s, z_{i_2}^s, \dots, z_{i_m}^s \\}$, where each $z_{i_j}^s := f(x_{i_j}^n)$ denotes the representation produced by the SSL pretrained model, and $g_n(X) := \\{z_{i_1}^c, z_{i_2}^c, \dots, z_{i_m}^c\\}$, where each $z_{i_j}^c := g_n(x_{i_j}^n)$ denotes the client's feature vector."
- page 4, last paragraph, last three lines: "This is achieved as follows: For each $x_{i_j}^n$ in $\\{x \_{i_j}^n \\} \_{j=1}^{m}$, consider $\\mathcal{N} = \\mathcal{N}(x \_{i_j}^n,K) \\subseteq \\{ i_k \\} \_{k=1}^m$ the set of indices of the $K$-nearest neighbors of $x_{i_j}^n$ in the representation space of the SSL model (where $K\leq m-1$), and create $K$ positive pairs $\\{(z_{i_j}^s, z_{i_k}^s)| i_k\in \mathcal{N}\\}$ and the remaining $m-K-1$ negative pairs $\\{(z_{i_j}^s, z_{i_k}^s)| i_k\in \mathcal{N}, i_k\neq i_j\\}$. Here, the $K$-nearest neighbors $z \_{i_k}^s$ are determined by the Euclidean distance between the normalizations of $z \_{i_j}^s$ and $z \_{i_k}^s$." Note in particular that there are multiple superscript typos in this paragraph: The superscript for all "$z$" variables should be "$s$" (for SSL representations), not "$c$".
- page 5, line before eqn (2): "$z \_{i_j}^s$", not "$z \_i^c$". In particular, the superscript should be "$s$".
- page 5, eqn (2): I would suggest the following:
$$\\mathcal{L} \_{CL,i_j}(X;f,g_n) := -\\log \\frac{\\sum_{i_k\in \\mathcal{N}} \\mathrm{exp}(\\langle z \_{i_j}^s, z \_{i_k}^s \\rangle / \\mathcal{T})}{\\sum_{l=1}^m \\mathrm{exp}(\\langle z \_{i_j}^s, z \_{i_l}^s \\rangle / \\mathcal{T})} $$

(In particular, the superscripts for all "$z$" variable should be "$s$".
Notation-wise, the authors should be $\\log$, not $log$. Also, the notation for inner product should be $\\langle x,x' \\rangle$, not $<x, x'>$ (i.e. use \langle, \rangle).)
- page 5, eqn (3): Here, the notation for the predicted logit clashes with the notation for the ground-truth label. I would suggest the following:
$$\\mathcal{L} \_{i_j}(x_{i_j}^n;X, \\hat{Y},f,g_n,h_n) := \\mathcal{L}_{CE}(\\hat{y} \_{i_j}^n, h_n(g \_n(x \_{i_j}^n))) + \\lambda \\cdot \\mathcal{L} \_{CL,i_j}(X;f,g_n). $$

- page 5, 1st line after eqn (3): "$\\mathcal{L} \_{CL, i_j}$", not "$L \_{CL}$". Also, with the updated notation, "$\\mathcal{L} \_{CE} (\\mathrm{y}_i, \\hat{y}_i;(g,h))$" should be replaced by "$\\mathcal{L} \_{CE}(\\hat{y} \_{i_j}, h_n(g \_n (x \_{i_j}^n)))$".
- page 5, 2nd line after eqn (3): The notation $h_c$ is undefined, and the sentence is unclear. I would suggest the following change: "... between the predicted logit $h_n(g \_n(x \_{i_j}))$ and the label $\\hat{y} \_{i_j}^n$, which involves the weights for both $g_n$ and $h_n$ of client $n$."
- page 5, 3rd line after eqn (3): "... the logits $h_n(g \_n(x \_{i_j}^n))$."
- page 5, 4th line after eqn (3): "$\\mathcal{L} = \\frac{1}{m} \\sum_{j=1}^m \\mathcal{L} \_{i_j}(x \_{i_j}^n; X, \\hat{Y}, f,g_n,h_n)$".
- page 5, 1st line before eqn (4): "... rewrite Equation 3 in terms of representations $z \_{i_j}^c$ and $z \_{i_j}^s$ as ..."
- page 5, eqn (4): There are several superscript typos in the second term. This equation should be
$$\\mathcal{L} \_{i_j}(x \_{i_j}^n; X, \\hat{Y},f,g_n,h_n) = \\mathcal{L} \_{CE}(\\hat{y} \_{i_j}^n, h_n(z \_{i_j}^c)) - \\lambda \\cdot \\log \\frac{\\sum_{i_k\\in \\mathcal{N}} \\mathrm{exp}(z \_{i_j}^s \\cdot z \_{i_k}^s / \\mathcal{T})}{\\sum_{l=1}^m \\mathrm{exp}(z \_{i_j}^s \\cdot z \_{i_l}^s / \\mathcal{T})}. $$
In particular, note that the superscript for the "$z$" variable in the first term is "$c$", while the superscripts for the "$z$" variables in the second term are "$s$".
- page 5, 3rd line after eqn (4): "$z \_{i_j}^s$", not "$z \_i^c$".
- page 5, 6th line after eqn (4): "$z \_{i_j}^s$", not "$z \_i^c$". Also, "$z \_{i_k}^s$", not "$z \_k^c \\in \\mathcal{N}$". (Note that $\\mathcal{N}$ contains indices $i_k$, and not representations $z \_{i_k}^s$.)
- page 11, Sec. 6, line 1: Technically, the approach to "locally regularize client models by leveraging the feature representations of self-supervised pretrained models" is not novel. The word "novel" in this sentence should be omitted.
- page 16, Algorithm 1: Notation should be updated accordingly, based on the above comments. (For example, "$\\hat{D} \_n = \\{(x \_i^n, \\hat{y} \_i^n)\\} \_{i=1}^{M_n}$", not "$\\hat{D} \_n = \\{(x_i, \\hat{y} \_i)\\}_{i=1}^{M_n}$".)

In addition, there are some (minor) grammatical errors/typos that should be corrected:
- page 2, paragraph before itemized list of contributions, 7th last line: "SSL-pretrained weights", not "an SSL pre-trained weight".
- page 2, Sec. 2, 1st paragraph: "... have been the core focus ...", not "... has been the core focus ..."
- page 4, 2nd line before eqn (1): "... sends the updated weights back to ..."
- page 5, 2nd line before eqn (3): "per-sample", not "per sample".
- page 17, 2nd last line: "Figure 11", not "11".
- page 19, 1st line: "Figures 12 and 13 present ..." (i.e. "Figures", not "Figure"; "present", not "presents").

**2) Additional related work on contrastive FL.** I also have one suggested change that would strengthen this paper. In the Related Work section (Sec. 2), it would be good to include related works on "Contrastive Federated Learning". Some relevant references include the following:

[1] Li, Qinbin, Bingsheng He, and Dawn Song. "Model-contrastive federated learning." Proceedings of the IEEE/CVF conference on computer vision and pattern recognition. 2021.

[2] Tan, Yue, et al. "Federated learning from pre-trained models: A contrastive learning approach." Advances in neural information processing systems 35 (2022): 19332-19344.

[3] Mu, Xutong, et al. "Fedproc: Prototypical contrastive federated learning on non-iid data." Future Generation Computer Systems 143 (2023): 93-104.

I would urge the authors to do a more comprehensive literature review on contrastive FL (beyond the above three suggested references).

---

> ### Author Response · Authors · 2025-09-01
> **Response to Reviewer 2uGV**
>
> We would like to thank the reviewer for their extensive suggestions on technical notations and grammar. We truly appreciate their effort. We have revised our manuscript as per your suggestions and included notational corrections in Section 3.0 and grammatical corrections throughout the paper. Below, we would like to make some clarifications:
>
> > In the experiments section (Sec. 4), the authors have explicitly stated that an existing code implementation of SimCLR (Chen et al. 2020) was used for their proposed contrastive regularization term. In particular, the contrastive loss objective used is the same as the contrastive loss objective from SimCLR.
>
> We apologize for this confusion. In Section 4 (Experiments), we outlined implementation details for our experiments. Particularly, we aimed to mention that the pretrained SSL model that we use for regularization was originally pretrained using SimCLR. We would like to clarify that while our objective is indeed derived from the InfoNCE and SimCLR objective, it differs in that we use multiple positives in the numerator while SimCLR uses a single positive example. For SimCLR, the positive example is the augmented version of the same input image, while in our scenario we utilize k-Nearest Neighbors identified using distances in the SSL representation space as the positive examples. We perform a hyperparameter study to validate the choice of the number of positives, $k$ in Figure 4, which we place here in the form of a table.
> | k     | Best Accuracy (%) |
> | ----- | ----------------- |
> | 1  | 78.70 ± 1.18      |
> | 2  | 79.22 ± 1.51      |
> | **4**  | **79.31 ± 1.43**      |
> | 8  | 78.63 ± 1.50      |
> | 14 | 78.38 ± 1.25      |
> | 24 | 75.34 ± 0.77      |
>
> *Table:* Hyperparameter study of k for CIFAR 10 (0.7, 0.5)
>
> We have rephrased our text in Section 4 for better clarity.
>
> > multiple superscript typos in this paragraph: The superscript for all "$z$" variables should be "$s$" (for SSL representations), not "c”.
>
> We apologize for the confusion. We would like to clarify that the k-contrastive objective is used directly between the classifier representations $z^c$. While we utilize the representations obtained from the SSL encoder to merely identify the indices for positive and negative pairings, the contrastive regularization objective is applied only on the classifier representations. Thus, the pretrained SSL encoders are kept frozen during the entire process. We have tried to clarify the confusion by making edits in the Methodology section (Section 3).
>
> >  I also have one suggested change that would strengthen this paper. In the Related Work section (Sec. 2), it would be good to include related works on "Contrastive Federated Learning".
>
> Thank you for the helpful suggestion. We have added a dedicated subsection in the Related Work section discussing relevant studies on Contrastive Federated Learning.

---

### Review · Reviewer_dmGE · 2025-08-20

**Summary Of Contributions:**

This paper proposes a local regularization method, "Local K-Similarity Constraint", to address the challenging problem of federated learning with noisy labels, particularly in scenarios where a majority of clients possess noisy data.

The core contribution is a technique by enforcing that data points semantically close in the SSL space remain close in the client model's representation space, thus mitigating the impact of incorrect labels.

Strengths:

The method demonstrates significant performance gains over existing approaches, especially in high-noise settings.


Weaknesses:

The method's performance is inherently dependent on the quality and domain-relevance of the pre-trained SSL model's representations.

**Audience:**

Yes

**Audience Explanation:**

Federated Learning / Learning with Noisy Labels Researchers.

**Broader Impact Concerns:**

It's good that there are real contributions, but I worry that they are too tiny for a publication in TMLR.

**Claims And Evidence:**

Yes

**Claims Explanation:**

The authors conduct experiments across six different datasets, which cover diverse domains and noise types.

**Requested Changes:**

It would be insightful to briefly discuss or hypothesize how performance might change if an SSL model pre-trained on a large-scale medical image dataset (if available) were used instead.

---

> ### Author Response · Authors · 2025-09-01
> **Response to Reviewer dmGE**
>
> We sincerely appreciate the reviewer's time and constructive suggestions. Below, we address raised concerns.
> >The method’s performance is inherently dependent on the quality and domain-relevance of the pretrained SSL model’s representation.
>
> Thank you for your feedback. We agree that the performance of our approach inevitably depends on the quality and domain of the pretrained SSL model’s representations. However, as we have mentioned in the Introduction, our core contribution is not in identifying the best SSL model for regularization, but in proposing a more effective strategy for leveraging the representations of readily available pretrained models.
>
> We would like to draw the reviewer’s attention to Table 2. Our experiments demonstrate strong cross-domain performance: even representations from a natural domain SSL model significantly improve robustness on medical datasets. Importantly, our method is architecture-agnostic, allowing any pretrained encoder that outputs representations to be seamlessly incorporated. This flexibility turns the dependence on model quality into a strength.
>
> For instance, Table 6 shows that substituting DINO for SimCLR, where DINO provides higher-quality pretrained features, yields substantial gains in robustness across both in-domain(natural) and cross-domain(medical) tasks. As the field increasingly produces higher-quality SSL models trained with better pretraining approaches and larger datasets, our framework becomes even more relevant, since it is designed to flexibly take advantage of such advances.
> |Method (Ours)|CIFAR10 (0.7, 0.5)|PathMNIST (1.0, 0.2)|
> |-|-|-|
> |w SimCLR (ResNet50)|79.31±1.43|93.58±0.26|
> |w DINO (ViT-S/16|86.34±0.13|94.02±0.16|
>
> *Table 6:* Study with pretrained ViT-S/16 DINO encoder.
>
> We have also added a section in Appendix A.2 to clarify our core contribution and discuss this point.
> >It would be insightful to briefly discuss or hypothesize how performance might change if an SSL model pre-trained on a large-scale medical image dataset (if available) were used instead.
>
> |Dataset|(0.7,0.2)|(0.7,0.5)|(0.85,0.2)|(0.85,0.5)|(1.0,0.2)|(1.0,0.5)|Avg %|
> |-|-|-|-|-|-|-|-|
> |CIFAR-100|46.44|58.94|58.18|77.47|64.25|106.62|68.65|
> |CIFAR-10 |15.31|23.65|24.96|37.53|38.45|55.03|32.48|
> |MURA|6.87|8.70|10.73|16.13|30.43|53.03|20.98|
> |PathMNIST|2.86|2.19|4.71|6.79|6.43|14.11|6.18|
>
> *Table:* Percentage robustness improvement of our method over FedAvg.
>
> Thank you for the thoughtful suggestion. Based on Table 2, we present a table comparing percentage gain in robustness across different datasets at various noise rates. Specifically, we calculate
>
> $$
> \frac{\text{Mean Performance(Ours)}-\text{Mean Performance(FedAvg)}}{\text{Mean Performance(FedAvg)}} \times 100%
> $$
>
> at each noise level for all datasets. We can infer that in-domain performance improvement(on natural datasets using natural SSL model) is larger than cross-domain performance improvement(on medical datasets using natural SSL model). Based on this, we hypothesize that an SSL model pretrained on a large-scale medical dataset could provide better representations for a more effective regularization against federated label noise with medical data.
>
> However, due to the current limited availability of large-scale medical pretraining datasets and models, natural image pretraining remains more practical and widely available. Our method emphasizes how such models can still be leveraged effectively in medical applications. With recent trends in medical image analysis ([1],[2],[3]) the availability of pretrained models is increasing, consequently the relevance of our method will also increase. We have added this discussion to Appendix A.2.
>
> [1] Ma, DongAo, et al. "A fully open AI foundation model applied to chest radiography." Nature 2025.
> [2] Chaurasia, Abadh K., et al. "A generalised vision transformer-based self-supervised model for diagnosing and grading prostate cancer using histological images." Prostate Cancer and Prostatic Diseases 2025.
> [3] Altaheri, Hamdi, et al. "Bridging Brain with Foundation Models through Self-Supervised Learning." arXiv:2506.16009 2025.
>
> >It’s good that there are real contributions but I worry that they are too tiny for a publication in TMLR.
>
> We politely disagree with the reviewer’s assessment. Our work explores the largely overlooked setting of high-noise, where existing baselines peform suboptimally. In addition, we have proposed a novel contrastive regularization objective for effective utilization of self-supervised pretrained models in federated label noise learning. Our objective is particulary suited for federated learning because of the architecture agnostic nature of the framework, allowing use of knowledge from large scale pretrained models with **no additional communication cost**. Our method also demonstrates strong performance, outperforming baselines by a considerable margin. We believe our contributions are substantial to the federated label noise research community.

---

### Decision · Action_Editor_vdUt · 2025-10-12

**Recommendation:** Reject

**Additional Comments:**

The contributions and evidence are not sufficient.

**Audience:**

Yes

**Audience Explanation:**

All reviewers agree that the topic of the manuscript has an audience. Considering label noise is definitely relevant to the federated learning subcommunity and the TMLR readership. Although, things such as convergence are usually topics of importance for the federated learning subcommunity.

**Claims And Evidence:**

No

**Claims Explanation:**

Not all reviewers agree that the manuscript presents appropriate evidence for its claims. The authors agree that the performance of their approach depends on the quality and domain of the pretrained SSL embeddings. Even after the authors' response, two reviewers remain unconvinced of the merits of the paper in light of the above. There are several ways to improve the manuscript, for instance, by including some analysis regarding training stability under mismatched SSL embeddings.

**Resubmission Of Major Revision:**

The authors may consider submitting a major revision at a later time.